# Assessing Compliance in Digital Advertising: A Deep Dive into Acceptable Ads Standard

## ABSTRACT

Online ads are a source of revenue for millions of websites. However, their intrusive and disruptive nature can impact the user experience of site visitors. Specialized tools such as browser extensions have emerged that block such advertisements from displaying. To restore balance in the favor of domain owners who lost revenue due to ad-filtering, online ad standards were defined to strike a middle ground between user choice and monetization. This paper presents a comprehensive analysis of the compliance of online digital advertisements with the most prevailing ad standard: the Acceptable Ads Standards. We selected 10,000 domains by intersecting Tranco's top 100K domains with the Acceptable Ads exception list. This subset highlights popular sites that are expected to adhere to specific advertising standards. The Acceptable Ads Standards, initiated by the Acceptable Ads Committee, seeks a balance between user experience and ad effectiveness, allowing certain non-intrusive ads defined by size, placement and type limitations. Our research methodology includes a quantitative analysis of ad formats and compliance rates. In this study, we conclude that almost 10% of the partner websites when crawled with Acceptable Ads' exception list have at-least one non-compliant ad on the landing page. Our analysis also reveals the design flaws in Acceptable Ads Exception list that allows publishers to bypass ad size and format limitations. Leveraging this understanding, we also propose improvements to the Exception list that can avoid violating ads from being rendered and ensure user experience of millions of site visitors who rely on Acceptable Ads is improved.

## CCS CONCEPTS

• **General and reference** → **Measurement**; *Validation.*

## KEYWORDS

Web Measurement, Digital Advertisement, Web Standard

## 1 INTRODUCTION

In today's digital age, the internet has become an integral part of our lives, with a significant portion of our time spent navigating through its vast and diverse content. This digital journey often involves encounters with a wide array of online advertisements, which play a crucial role in the economic framework of the web. Advertisements online take various forms, including, but not limited to, display ads such as banners and pop-ups, video ads that often play before or during accessing online content, native ads that blend seamlessly with the content of the webpage, and interstitial ads that appear between page transitions.

While these advertisements are essential for keeping many websites operational and content freely accessible, their disruptive nature has raised concerns. Intrusive ads can lead to negative user experiences, prompting the development of ad standard aimed at improving the web environment. The Better Ads Standards and Acceptable Ads Standard emerged as key guidelines in this respect [5, 7]. The Better Ads Standards, initiated by the Coalition for Better Ads, target eliminating ads deemed excessively intrusive or bothersome to users. On the other hand, the Acceptable Ads Standard, guided by the Acceptable Ads Committee, strives to find a middle ground that allows for ads which are non-intrusive and acceptable to users, thereby ensuring that websites remain profitable without compromising the user experience.

This paper delves into the current landscape of online advertising, with a particular focus on the compliance of these ads with Acceptable Ads Standard. We focus on the Acceptable Ads Standard as it is the *default advertising policy* for many ad blockers, such as Adblock Plus, which allows certain non-intrusive ads to be shown. As this standard is automatically applied by default, it impacts around 300 million global users [13]. Our goal is to assess how effectively the Acceptable Ads Standard committee oversees the exception rules that permit ads to be displayed under this framework. We note that while other works have explored the compliance of other ad standards like Better Ads Standard [45], we are the first to assess the compliance of Acceptable Ads Standard, which is generally considered more rigorous in their guidelines for allowed ad formats [1]. Our work seeks answer to the following research questions to better understand the compliance of acceptable ads.

**RQ1:** *Are there non-compliant ads on partner websites exempted under the Acceptable Ads Standard? If so, how prevalent are they?* We examine online ads on domains that are exempted from ad-blocking under Acceptable Ad's Standard. Additionally, we assessed the role of various ad publishers in contributing to these violations. Our study uncovers patterns of non-compliance and identifies major offenders. From our analysis of Tranco's top 100K websites that include exception rules for acceptable ads, we found that approximately 10% of these sites display at least one ad that fails to meet compliance standard.

**RQ2:** *What flaws and limitations exist in the current exception list that contribute to the prevalence of violating ads?* We utilize our telemetry data of violating ad elements from the web and evaluate the overly permissive rule structures from the exception list to find limitations in the current enforcement of acceptable ads.

**RQ3:** *Can the exception list endorsed by the Acceptable Ad committee be enhanced to reduce the non-compliance rate of*

*Conference acronym 'XX, June 03–05, 2018, Woodstock, NY*
© 2018 Copyright held by the owner/author(s). Publication rights licensed to ACM.
ACM ISBN 978-1-4503-XXXX-X/18/06…$15.00
https://doi.org/XXXXXXX.XXXXXXX

*violating ads on partner websites?* Based on our findings regarding the limitations and flaws of allowlist rules, we propose ways to enhance the enforcement of acceptable ads by implementing more precise allow rules and avoiding overly permissive ones. We evaluate our proposed enhancement by demonstrating reduced non-compliant ads when testing on real-world websites.

In summary, we make the following contributions to enhance the understanding and improvement of ad compliance on websites adhering to the Acceptable Ads Standard:

- We developed a web crawling tool that utilizes a proxy-based approach to inject scripts into web pages for measuring properties of web elements. Additionally, we crawl the same page using different configurations of ad filtering rules, enabling us to retrieve the ads of interest. Additionally, the injected script performs in-situ telemetry to identify ad elements that violate Acceptable Ads Standard. We will also open-source our measurement framework to the public.
- We conducted a comprehensive web measurement study involving 10,000 domains selected from the intersection of Tranco's 100K domains and partner websites listed in the Acceptable Ads exception list. Our findings indicate that one in every ten websites display at least one violating ad.
- Leveraging the telemetry data collected, we identified overly permissive rules and DOM elements consistently associated with violating ads. We implemented improvements to the exception list and demonstrated a reduction in violating partner websites by 32.4%.

These contributions enhance the existing literature on ad compliance and provide actionable insights for refining ad standards.

## 2 BACKGROUND

Online advertising is a primary revenue model for millions of websites, providing financial support for free content and services. In the U.S alone, the online advertising market surpassed 225 billion dollars in 2023 [16]. Digital advertisements come in various forms, including text ads, video ads, pop-ups, and in-video ads. Despite their economic significance, advertisements can often hinder user experience due to their disruptive nature, leading to the development of ad-filtering technologies.

Ad-filtering software, or ad blockers, have become widely popular as users increasingly seek to minimize interruptions and protect their privacy. These tools block or hide unwanted ads from being displayed on websites, offering a cleaner browsing experience. However, the widespread adoption of ad blockers poses challenges for websites that rely on ad revenue, prompting the development of acceptable ad standards and efforts to strike a balance between ad-based revenue and user control.

### 2.1 Ad-Filtering and EasyList

A cornerstone of ad-filtering is the use of blocklists, which define the specific rules for identifying and blocking ads. One of the most prominent of these blocklists is *EasyList* [11]. EasyList is an open-source, community-maintained list that contains rules used by most ad blockers to filter unwanted ads. These rules cover a wide range of ad types, including banners, pop-ups, and tracking elements. EasyList, integrated into popular ad-blockers like AdBlock Plus [3]

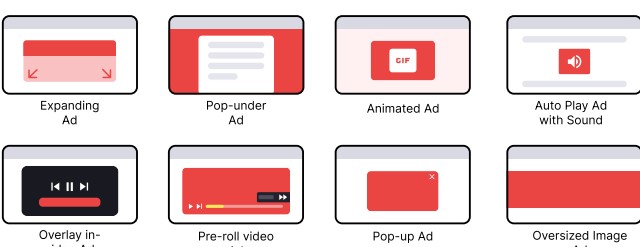

Figure 1: Ad formats strictly forbidden under acceptable ads.

and uBlock Origin [20], provides rules for blocking or hiding URL patterns, CSS selectors, and scripts, enabling seamless filtering of intrusive elements from webpages.

The list is continually updated by a group of volunteers and contributors who review and add new rules based on user submissions or as new advertising techniques emerge. EasyList also contains regional variations, known as supplementary lists, to accommodate language- and region-specific ads. While EasyList is highly effective at reducing intrusive advertisements, it can hurt the economic perspectives of domain owners who may rely on ad monetization. This has led to efforts to develop standards for non-intrusive ads that meet both user and publisher needs.

### 2.2 Acceptable Ad Standard

The Acceptable Ads Standard aims to balance user experience with website monetization by allowing certain non-intrusive ads that are less disruptive. It creates an exemption for non-intrusive ads by declaring rules in an Exception list [4] that is structured similarly to the Easylist. The standard describes in detail the distinction, size, and placement of the ads in the following manner:

- **Size:** Ads must occupy a reasonable amount of screen space, with specific size and dimension limitations.
- **Placement:** Ads should be clearly distinguishable from the primary content.
- **Labeling:** All ads must be clearly labeled as such.

In contrast, the following types of ads, shown in Figure 1, are deemed unacceptable and are considered violations of the standard:

- **Pop-ups and Pop-under Ads:** Ads that appear in separate frames, windows or tabs, either above or below the current context.
- **Animated Ads:** Advertisements with rapid animations or flashing effects.
- **Audio/Video Ads with Sound:** Advertisements that play audio or video with sound automatically upon loading of the page.
- **Ads Covering Content:** Ads that cover significant portions of the webpage's content.

These criteria formed the foundation for our heuristics to automate ad vetting, ensuring that only ads conforming to the Acceptable Ads Standard are allowed. The heuristics, which we discuss in Section 4.3, were tailored specifically to desktop ads, with mobile ads being outside the scope of this analysis.

## 3 RELATED WORK

**Online Ads**. The subject of digital advertising has attracted significant attention from various stakeholders within the online ecosystem [24, 30, 42]. Economic incentives have driven research into the

effectiveness of different advertising formats and the key factors that capture user attention [28, 36, 38]. Some studies approach this issue from a privacy perspective, highlighting the potential harms posed by targeted advertising to users' privacy [23, 26]. These privacy concerns have fostered the development of tools and privacy controls designed to empower users to defend against tracking by advertising entities [33, 37]. Additionally, security researchers have exposed vulnerabilities in ad systems, demonstrating how they can be exploited to perpetrate fraud against online users [22, 35]. For example, Oentaryo et al. [34] outlines methods for detecting fraudulent ad publishers who generate deceptive ad links aimed at misleading users.

**Ad Blocking.** Prior research has extensively examined various aspects of online ad experiences, with particular attention to intrusive ads. In response, several tools and extensions have been developed for ad blocking, such as Adblock Plus [3], uBlock Origin [20], and Ghostery [2]. These tools primarily rely on community-maintained blocking lists, like EasyList (for ads) [11] and EasyPrivacy (for trackers) [12], to block specific content URLs. Additionally, some studies have explored automated approaches, such as machine learning classifiers, to adapt to evolving ad and tracker characteristics [29, 32]. These solutions aim to block all types of ads.

**Ad Compliance.** The issue of ad compliance has become increasingly prominent as it pertains to the quality of online ad experiences. Early on, governmental organizations such as the FTC established guidelines to promote greater transparency among websites and publishers within the digital advertising ecosystem [25]. These governmental frameworks have spurred the creation of self-regulatory bodies by ad publishers to ensure compliance with industry standards [9, 15]. Various studies have evaluated publisher adherence to organizations such as NAI and DAA [31]. More recently, regulations focused on user data, including GDPR and CCPA, have had a profound impact on the digital advertising landscape [8, 27]. Research has examined the effects of these data protection regulations on advertising practices [39, 41], with findings showing that despite such regulations, ad publishers continue to adapt their methods to collect user data for targeted advertising.

In addition to these regulatory frameworks, ad policies like the Acceptable Ads Standard [5] and the Better Ads Standard [7] have provided explicit guidance on ad practices, such as size, placement, and display rules, to minimize disruption to users while allowing site owners to earn through ad monetization. Researchers have studied the impact and privacy implications of these ad policies [40, 47], and Yan et al. have quantitatively assessed the effectiveness of the Better Ads Standards [45]. To our knowledge, however, we are the first to conduct a detailed examination of compliance with the Acceptable Ads Standard by partner websites and publishers.

## 4 METHODOLOGY

This section outlines the methodology we deploy to crawl the webpages for identifying ad types. It discusses the technical details of our crawler, including the various configurations we use to discover ads. Lastly, it discusses the design of our heuristics for non-compliant ads that help us determine the compliance rates.

### 4.1 Ad Filtering Configurations

We utilize the functionality of AdBlock Plus [3] to block/allow ads on the webpage. The extension allows configuring various blocking and allow lists. For our approach, we develop three configurations that are important to the two-crawl process:

- $C_{Ads}$: The first crawl does not use any list. The second crawl uses only EasyList (i.e., all ads are blocked). The difference gives us all ads. We will refer to this dataset of ads as $\mathcal{D}_{Ads}$, and use it to report the overall prevalence of ads obsserved on the web.
- $C_{acceptableAds}$: The first crawl uses EasyList and Exception list and the second crawl uses only EasyList. The difference in the elements recorded in the two crawls gives us acceptable ads only.[1] We will refer to this dataset of Acceptable Ads as $\mathcal{D}_{acceptableAds}$ and use it to report the frequency of forbidden ad types observed despite the Acceptable Ads Standard enforcement through an exception list.
- $C_{modified}$: In this study, we also propose solutions to improve the Exception list. To test the impact of these changes, we modify the Exception list and implement the following configuration: The first crawl uses EasyList combined with the modified Exception list, while the second crawl uses only EasyList. By comparing the elements captured in both crawls, we identify the acceptable ads displayed after filtering out non-compliant ads. We will refer to this dataset of acceptable ads as $\mathcal{D}_{modified}$ and use it to demonstrate the effectiveness of our proposed changes in improving the Exception list that provides a greater degree of adherence to the Acceptable Ads Standard.

### 4.2 Two-Crawl Detection Approach

Our configurations enable the tool to capture the required ads based on the configuration we use. Below, we outline the tool's workflow as it crawls webpages and collects telemetry data.

For each domain, the tool performs two consecutive crawls with a 10-second delay between them, ensuring minimal changes to the webpage within this time frame [21]. During each crawl, we inject a script into the webpage's head using mitmproxy [14], with the defer attribute set to run before the DOMContentLoaded event. This script scans the page and lists all elements, including media content like images, videos, and SVG files. Once all resources (scripts, images, subdocuments, etc.) are loaded (triggering Load event), the script waits 10 seconds to ensure rendering completion before traversing the DOM. If the Load event doesn't trigger, the tool stays on the page for up to 60 seconds before terminating. If loading fails, we retry the domain once before removing it from analysis.

Since frames are isolated by same-origin-policy, we use mitmproxy to modify Cross-Origin Resource Sharing (CORS) flags and configure the browser to disable web security. This ensures the script can access all resources loaded in the browser.

The script captures details such as CSS properties, class names, XPaths, optimized XPaths, and other attributes of each element. It is important to note that ad resources are typically placed in well-defined sections of the DOM, and repeated visits to the webpage will render ads in the same locations. Ad-blocking tools utilize this deterministic behavior to hide the DIV elements assigned to

---

[1]Acceptable Ads are a subset of generic Ads found on the web.

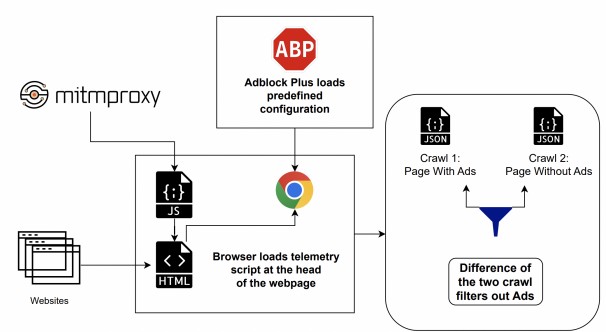

**Figure 2: Two-crawl ad detection. `Mitmproxy` injects a scanner script that traverses DOM object. Difference of the two crawl reveals ads.**

ad networks. We also leverage this determinism to identify ads wherever they appear on the rendered webpage. By comparing the lists of web elements generated from the two crawls (using XPaths), we can identify the content blocked by AdBlock Plus and determine which elements were detected as ads (as shown in Figure 2).

### 4.3 Detection of Non-Compliant Ad Categories

We note that Better Ad Standards and Acceptable Ads Standard both block some common ad format (e.g. popup, popunders, autoplay media, interstitial and overlay). Therefore, we take inspiration from the work of Yan et al. (that has studied the ad types not allowed under Better Ads Standard) [45] to design heuristics for forbidden ad types in the Acceptable Ads Standard. We define heuristics for six specific types of non-compliant ads based on their CSS properties. These specific formats were chosen due to their clear violation definitions and detectability, as demonstrated by existing work [45]. These include:

- **Over-Sized Image Ads**: While The Acceptable Ads Standard prohibits 'Generally Oversized Image' ads but does not specify exact size limitations. For our analysis, we consider ads occupying more than 80% of the screen's width or height blatantly oversized.
- **Autoplay Media**: Media ads are identified if they have the `autoplay` attribute enabled or are automatically preloaded. This applies to both videos and images
- **Overlay Ads**: There are ads with sticky ads that are placed on the top or at the edge of the viewport and have `fixed` or `absolute` positioning
- **Popup Ads**: Popups are detected by their high `z-index` combined with `fixed` or `absolute` positioning
- **Popunder Ads**: Popunders are similar to popups but are positioned beneath content, indicated by a negative `z-index`
- **Interstitial Ads**: These are full-screen ads detected if they cover more than 75% of the viewport and have `fixed` or `absolute` positioning

These rules enable the systematic identification of different types of non-compliant ads across crawled domains.

### 5 DATA COLLECTION

To collect data for measuring online advertisements and their compliance with Acceptable Ads Standard, we developed a specialized

**Table 1: Number of partner domains per rank division analyzed in our measurement.**

| Rank Division | Domain Count |
|---|---|
| 1-1,000 | 326 |
| 1,001-10,000 | 1,791 |
| 10,001-100,000 | 7,883 |

tool, the details for which have been described in Section 4.2. This section focuses on the process and setup used for data collection, including the selection of websites and the technical infrastructure used for crawling.

### 5.1 Website Selection

We aimed to analyze advertisements on a broad and diverse set of domains. To achieve this, we chose to crawl websites that are in the intersection of two specific lists: the Tranco top 100K websites [19] and the first-party domains found in the Acceptable Ads Standard's Exception list [4]. The Tranco list is a frequently updated ranking of the most popular websites on the internet, ensuring that our dataset reflects domains with significant user traffic. The exception list, on the other hand, contains rules for allowing ads on certain partner domains, provided they comply with Acceptable Ads Standard.

Table 1 shows the number of partner domains and their corresponding rank divisions. By selecting domains present in both the Tranco top 100K and the acceptable ad exception list, we ensured that our dataset included high-traffic websites that serve ads and are subject to compliance regulations. After excluding inactive or publicly inaccessible domains, we were left with a set of 9,463 domains for our analysis.

### 5.2 Crawling Setup

The data collection process was conducted on a server with 32 cores and 64 GB of RAM, enabling us to run 30 parallel crawling processes to expedite the data collection workflow. Each crawling process was tasked with visiting the selected domains and capturing the ad content displayed on the webpages. For automation, we used Puppeteer [17], a Node.js library that provides a high-level API for controlling headless browsers. To avoid detection by bot-detection algorithms, we employed Puppeteer's stealth plugin [18]. Additionally, we incorporated randomization in the scrolling behavior during each crawl to mimic human interaction patterns (e.g., each iteration of scroll-up and scroll-down had a random factor of movement), further evading potential bot detection.

### 5.3 Crawling Process

We restricted our crawling to the landing page of each domain. We detect failure to load a webpage by inspecting `mitmdump` file generated by the proxy. The dump file contains network exchange records and can be used to determine if the site server failed to respond. In that case, we simply reattempt one more time.

The browser remained on the page until the `Load` event is triggered, signaling that the initial content had fully loaded. To ensure all dynamically-rendered content, including ads, was captured, we allowed an additional 10 seconds of idle time. Additionally, we took screenshots of the web interaction before closing the browser, which were later used to report the violations.

Table 2: Counts of forbidden ad types found using $C_{\text{acceptableAds}}$ and $C_{\text{Ads}}$.

| Ad Types | $\mathcal{D}_{\text{acceptableAds}}$ | $\mathcal{D}_{\text{Ads}}$ |
|---|---:|---:|
| Oversized Ads | 3,410 | 22,878 |
| Autoplay Media | 20 | 297 |
| Overlay Ads | 3,865 | 23,891 |
| Interstitial Ads | 121 | 3,503 |
| Popup Ads | 465 | 2,529 |
| Popunder Ads | 29 | 358 |
| **Total** | **7,910** | **53,456** |

Crawling 9,463 domains in our setup took approximately two days to complete. The data was collected in the United States to ensure consistency in the results and avoid regional differences in ad serving practices. By utilizing parallel processes and a carefully curated domain set, our crawling infrastructure enabled efficient and thorough data collection, allowing us to evaluate ad compliance across high-traffic websites effectively.

## 5.4 Ethical Considerations

In our study, we are acutely aware of the ethical concerns surrounding web measurement and data collection. Our method of web crawling involves injecting a telemetry script into the web page, but it is essential to emphasize that this process does not alter or manipulate the webpage in any way. We do not generate additional requests to website servers; instead, our telemetry consists of inspecting web elements present on the page. The functionalities of our approach are akin to what can be achieved using browser developer tools, such as Chrome Dev tools. To minimize our impact, we remain on each site for a duration of no more than 60 seconds before closing the browser window. This approach ensures that our activities do not impose any load on the website's server, effectively preventing the risk of DDOS attacks or similar disruptions.

## 6 PREVALENCE OF VIOLATING ADS

This section will discuss our analysis on non-compliant ads found in our datasets, thereby answering **RQ1**: What is the prevalence of non-compliant advertisements on partner websites exempted under the Acceptable Ads Standard?

### 6.1 Effectiveness of Acceptable Ads Standard

As a first step towards understanding the prevalence of non-compliant ads on the web, we selected 10K domains that are common in Tranco's 100K domain set and Acceptable Ads Standard's Exception List. The latter list provides exception filter rules enforced on specific domains to unblock ads that are supposed to be compliant with the stated acceptable ad standard.

Our analysis focused on six specific categories of non-compliant ads, derived from the Acceptable Ads Standard. These categories, along with the number of instances detected across all domains, are summarized in Table 2. We highlight the frequencies found in the dataset $\mathcal{D}_{\text{acceptableAds}}$. For comparison, we also show the number of ads detected in $\mathcal{D}_{\text{Ads}}$. The contrast of the two configuration shows the prevalence of ads not conforming to the Acceptable Ads Standard. For instance, we observed that the clean browser profile

encounter 4.86 times more overlay ads than a browser profile with adblocking functionality. A Chi-Squared test [43] comparing ad type distributions between the configurations with and without Adblock Plus resulted in a highly significant difference ($\chi^2$ = 1907.24, $p < 0.0001$), indicating the two configurations have statistically different ad type distributions. Specifically, ad types such as oversized images, overlay ads, and interstitial ads show substantial differences, confirming that Adblock Plus effectively blocks or reduces certain types of ads more than others.

While the EasyList in combination with the Exception list is effective in significantly reducing intrusive/disruptive ads, it does not achieve complete blocking of such non-conforming ads. Our study found that 9.91% of the websites in our dataset displayed at least one violating ad among those deemed acceptable. In an ideal scenario, with full adherence to the Acceptable Ads Standard, $C_{\text{acceptableAds}}$ configuration should block all ads that violate these guidelines. However, our results suggest that the Exception list fails to fully enforce compliance, allowing ads to be displayed that do not meet the size and type restrictions it promotes.

> **Finding 1**: *Despite the effectiveness of EasyList in combination with the Exception list at reducing non-compliant ads significantly, we found violating ads to be present in 9.91% of the domains that were partners in Acceptable Ads Standard's exception list.*

Among the 9.91% of domains displaying violating ads, we analyzed the breakdown of the types of violations. Figure 3 shows the distribution of these categories, with each data peak representing the frequency of a particular ad type observed on a domain. The most common violations were Oversized Image Ads and Overlay Ads. Notably, certain domains exhibited higher frequencies of violations. For instance, `naszemiasto.pl` displayed 103 Overlay Ads, despite filtering under the Acceptable Ads Standard. This domain features continuous scrolling that feeds in overlay and oversized ads, trigger the heuristics to log all the elements found in Section 4.3. Another high-ranking domain, `express.co.uk`, displayed oversized ads below the primary content, violating size limitations. Autoplay Ads were the least frequent type of violation, though they were observed on some high-traffic sites, such as `gsmarena.com`. Appendix A contains example screenshots that highlight some of these violations.

> **Finding 2**: *Oversized Image Ads and Overlay Ads were observed to be the most common non-compliant ads in our dataset.*

### 6.2 Non-compliant Ad Publishers

We also assess the contribution of various ad publishers who display violating ads in $\mathcal{D}_{\text{acceptableAds}}$. As discussed in Section 4.2, our crawl gathers CSS properties of ad elements. For more complex cases, we also navigate through parent nodes, collecting class names and other CSS selectors if available. This method provides a rich metadata inventory for each ad, which can be used to trace the ad publishers that displays the ad.

Additionally, we match this data with filter rules from the exception list to identify the publisher responsible for showing the ad. Figure 4 shows an example of a metadata report generated

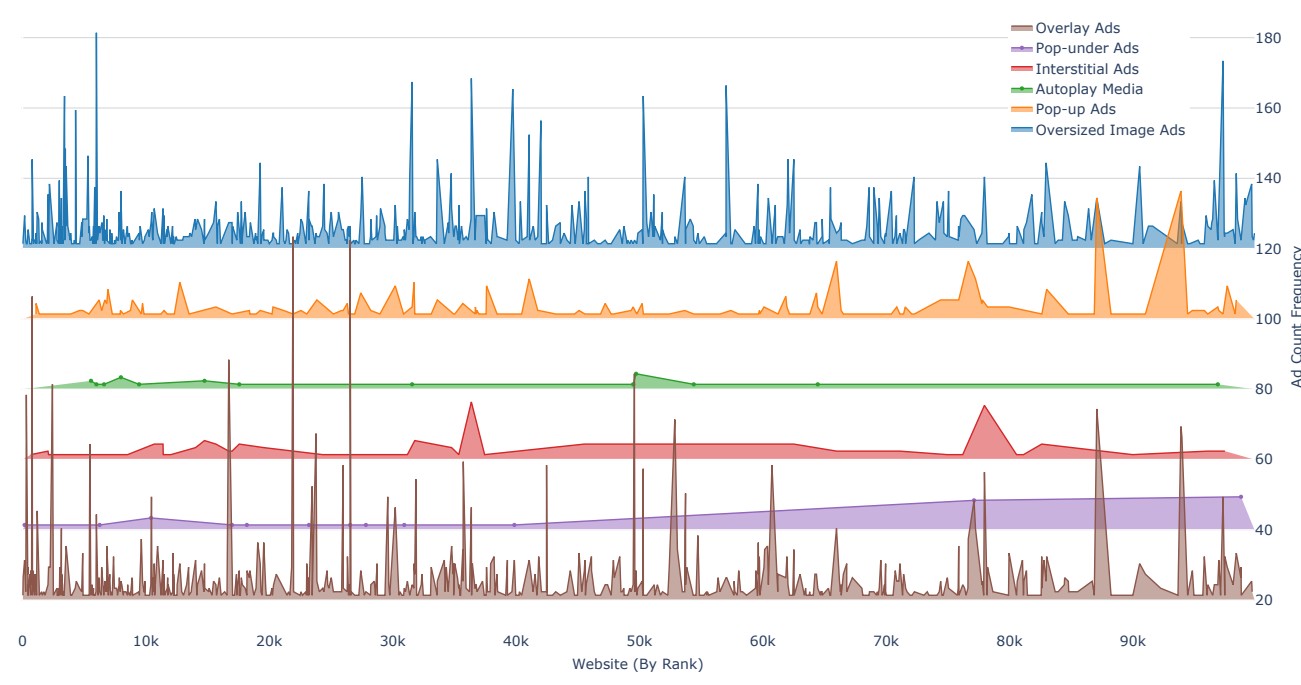

**Figure 3: Distribution of violating ads under Acceptable Ads Standard across the different ranked websites. Violations appear across all ranks.**

after crawling `gsmarena.com`. It highlights metadata from an overlay ad, which violates the Acceptable Ads Standard, published by `brwsrfrm.com`. The `filter_rule` refers to the exception list rule that allows this ad. In this instance, any ad element within a `DIV` with the classname `ad_label` is permitted, even if it violates other properties, such as triggering sticky or overlay behaviors.

```
{
    "ad_src": '/d/img/1108/fd3b2628-68dc-4bab-94a8-
        d6caa14bd2bf/14398?bid=0&w=300&h=600'
    "violation": ['OVERLAY'],
    'tag': 'IMG',
    'parent_tag': 'DIV',
    'parent_class': 'ad-label'
    'parent_src': 'https://brwsrfrm.com/d/if/.../14398?bid
        =0&w=300&h=600/.../'
    'filter_rule': 'gsmarena.com,aternos.org#@#.ad-label'
}
```

**Figure 4: Example of metadata of an Overlay ad found on `gsmarena.com`**

From this example, we can conclude that `brwsrfrm.com` is an ad publisher associated with a non-compliant ad instance. Our inventory of non-compliant ads identified during our crawling revealed multiple other publishers displaying invasive ads in a similar manner. To determine which ad publishers frequently displayed these violations, we utilized our metadata inventory to identify the parent companies that own these networks. First, we extract the URL of the ad element itself. If the URL is absent, we fetch the container

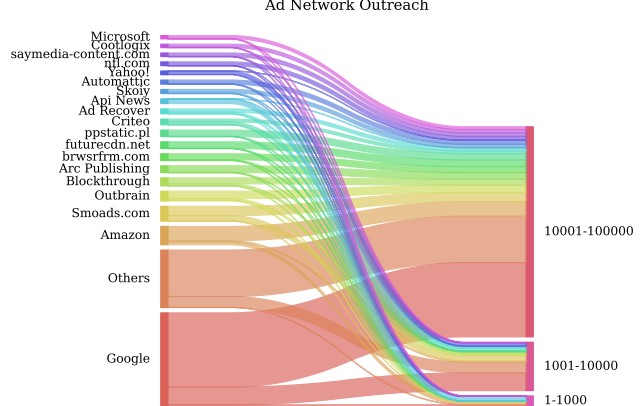

**Figure 5: Different ad networks' role in the distribution of non-compliant ads.**

frame's URL. Then, we extract the tld+1 (Top-level domain + 1) out of the URL. Lastly, we match these domains with entity and services domains found in the list of ad companies, along with the respective services under them, curated by Disconnect [10]. Figure 5 illustrates the share of ad publishers contributing to the prevalence of non-compliant ads across the three ranges of domain ranks we crawled. The top five ad publishing companies displaying violating ads are `Google`, `Amazon`, `Smoads.com`, `Outbrain`, and `Blockthrough`. This analysis underscores the role of various ad networks in perpetuating non-compliant advertising practices.

Table 3: Counts of non-compliant ads using $C_{\text{acceptableAds}}$ at two distinct time stamps.

| Ad Type | 09/7/2024 | 10/7/2024 | Change (%) |
|---|---|---|---|
| Oversized Image | 3,410 | 3,203 | 6.07% ↓ |
| Autoplay Media | 20 | 19 | 5.0% ↓ |
| Overlay Ads | 3,865 | 3,593 | 7.03% ↓ |
| Interstitial Ads | 121 | 105 | 13.2% ↓ |
| Popup Ads | 465 | 440 | 5.37% ↓ |
| Popunder Ads | 29 | 28 | 3.44% ↓ |

**Finding 3**: *The top five ad publishing companies that had displayed violating ads are Google, Amazon, Smoads.com, Outbrain and Blockthrough.*

## 6.3 Temporal Consistency of Violations

The presence and nature of ads on the web are subject to significant fluctuations due to various dynamic factors. Live global events, time of day, regional politics, and evolving user interests all influence the ads displayed. Additionally, ad publishers engage in real-time bidding [46], competing for ad space, which can further impact what users see at any given moment. Due to these variables, performing a second round of crawl for all the domains is crucial for assessing the consistency of ad violations. This approach ensures a more accurate evaluation of compliance over time, accounting for the changing landscape of digital advertising.

To evaluate consistency of violating ads, we have two rounds of crawl conducted on September 7, 2024 (Crawl 1) and October 7, 2024 (Crawl 2) using $C_{\text{acceptableAds}}$ configuration. We compare the proportions of various ad types found to be in violation. It was observed that these proportions of violating ad types in both crawls were quite similar. For instance, oversized images represented 43.11% (3,410) of violations in Crawl 1 and 43.61% (3,203) in Crawl 2. Overlay ads made up 48.86% (3,865) of violations in Crawl 1, while accounting for 48.38% (3,593) in Crawl 2. The differences across other ad types were similarly minor. Additional details on these discrepancies can be found in Table 3.

A Chi-squared test was conducted to assess the statistical significance of differences in ad distributions, yielding a statistic of $\chi^2 = 1.9432$ with 5 degrees of freedom [2] ($p = 0.8569$), indicating no significant difference. The expected and observed frequencies for each ad type closely aligned, suggesting consistency between the two crawls. Additionally, the Kullback-Leibler divergence [44] of 0.000259 further supports minimal divergence, reinforcing that the distribution of violating ad types remained consistent between the two time periods.

**Finding 4**: *The results of the temporal analysis indicate no significant differences in the distribution of violating ad types between the two crawls.*

## 7 IMPROVING ACCEPTABLE ADS STANDARD

The goal of this section is to identify the underlying reasons why non-compliant ads are displayed and discover ways to refine the

---

[2]Since we have 6 ad categories, the degrees of freedom is 5.

Exception list by eliminating problematic rules that allow these ads to interfere with the user experience. We accomplish this by parsing the list and analyzing the telemetry data collected during our crawl of the offending ads.

### 7.1 Primary Causes for Non Compliant Ads

We provide the primary causes that lead to the display of non-compliant ads, thereby addressing **RQ2**: What limitations and flaws currently exist in the Exception list? To answer this, we inspect the potential sources of violating ads in our dataset.

**Over-Permissive Rule**. We parse the Exception list and find rules that match with the violating domains in $C_{\text{acceptableAds}}$. For certain domains, we identified that `^$document` unblocking rules were being enforced. These rules effectively creating a complete exception for the specified domain by overriding any adblocking rule on the entire site [6]. Among the 9.91% of domains displaying non-compliant ads, 52 of them were found to have this `^$document` allowlisting rule in place, which represents 5.34% of the non-compliant domains. This high prevalence indicates a significant flaw in the current enforcement mechanisms, as such unrestricted allowlisting can lead to the display of invasive ads.

We contend that such rules contradict the intent of the Acceptable Ads Standard, which aims to strike a balance between user experience and monetization.

**Finding 5**: *Among the 9.91% of domains exhibiting non-compliant ads, 52 domains utilized the overly permissive* `document` *allowlisting rule.*

**Offending Element Unblocking Rule.** We also identify the container elements where non-compliant ads are rendered. These containers are detected and unblocked by their `classNames` using Exception list rules. Figure 4 shows an example of the metadata of violating ads, compiled during and after the crawl. This metadata includes the class and tag type of the parent node where the ad is embedded. From both crawls (09/07 and 10/07), we observe that these parent nodes consistently display the same violating ad format, regardless of the ad publisher, across multiple site visits. To mitigate the impact of violating elements, we gather all relevant rules that correspond to these elements in our dataset. 6 highlights some examples of rules within the Exception list that unblock offending content.

```
speedtest.net#@#.ads-right
cnn.com#@#.stack__ads
@@||teva.com^$document
knowyourmeme.com#@#.ad-unit-wrapper
pagesix.com,decider.com,nypost.com#@#.billboard-ad
```

Figure 6: Examples of exception rules that are removed from the Exception list.

### 7.2 Improving Compliance

In this section, we seek to answer **RQ3**: Can we improve the Exception list to enhance ad compliance rates. Based on our observation from Section 7.1, we identify all rules in the Exception list that are either overly permissive or that allow exceptions for parent containers displaying violating ads consistently. It should be noted

**Table 4: Counts of violating domains and forbidden ad types found using $C_{\text{acceptableAds}}$ and $C_{\text{modified}}$**

| Category | $\mathcal{D}_{\text{acceptableAds}}$ | $\mathcal{D}_{\text{modified}}$ | Change ($\Delta$) |
|---|---|---|---|
| Violating Domains | 937 | 634 | 32.4% ↓ |
| Oversized Image | 3,410 | 2,636 | 22.7% ↓ |
| Autoplay Media | 20 | 19 | 5.0% ↓ |
| Overlay Ads | 3,865 | 2,864 | 25.9% ↓ |
| Interstitial Ads | 121 | 122 | 0.83% ↑ |
| Popup Ads | 465 | 458 | 1.51% ↓ |
| Popunder Ads | 29 | 28 | 3.44% ↓ |

that we remove only those rules enabled on domains where the offending containers were found, ensuring that other domains are not affected.

Based on these new changes, we prepare a new configuration, $C_{\text{modified}}$, that includes the modified Exception list. We configure the crawler with this new setting to measure the occurrence of the violating ad type by crawling all the 937 offending domains found in $\mathcal{D}_{\text{acceptableAds}}$. Table 4 summarizes the changes in violating ad types. Notably, the total number of violating domains decreased significantly by 32.4%.

The results reveal overall reductions in the counts of ad types, particularly oversized images, and overlay ads, which saw decreases of 22.7% and 25.9%, respectively. In contrast, interstitial ads experienced a slight increase of 0.83%, likely due to the fluctuations of this ad type. Additionally, only marginal changes were noted in the counts of autoplay media, popup ads, and popunder ads. These observations suggest that the removal of CSS class identifier exemptions and the `^$document` allowlisting rules to predominantly impact oversized and overlay ads.

Overall, these findings highlight the effectiveness of refining the exception list in reducing non-compliance and improving the user experience. Therefore, we recommend the removal of these overly permissive and offending-element unblocking rules from the Exception list.

> **Finding 6**: *Improving Exception list by removing offending element-hiding and overly permissive* `^$document` *rules predominantly impacts oversized image and overlay ads.*

## 8 DISCUSSION

This study was driven by the heavy reliance of approximately 300 million users on the Acceptable Ads Standard, which promises a browsing experience free from invasive ads. The framework's standards are essential for enforcing accountability among partner domains and ad publishers, striking a balance between protecting the user experience and allowing domain owners to generate revenue through advertising. Despite the existence of these policies and a well-defined Exception list, our findings reveal that non-compliant ads remain prevalent. Specifically, our analysis shows that one in ten partner websites still displays at least one type of violating ad. The violations include oversized images, autoplay media, overlay ads, interstitial ads, and popups—all of which degrade the user experience that the Acceptable Ads Standard aims to safeguard.

These results highlight gaps in enforcement and suggest the need for stronger measures to ensure compliance.

To address these issues, we introduced improvements to the Exception list by identifying and removing rules that are either overly permissive or that unblock DOM elements that constantly contain the violating ads. This initiative represents a critical first step in ensuring that the standards used by millions can be iteratively refined and enhanced. Our telemetry data provides actionable insights that can guide future modifications to the exception list, ensuring a more compliant advertising ecosystem toward greater user experience.

**Ethical Disclosure.** We have reported our findings of non-compliance to Adblock Plus and Acceptable Ads Committee and are now awaiting their response. The report has included screenshots along with the rules that were triggered to generate the violating ads.

**Limitations and Future Work.** Our work marks an important first step in assessing the enforcement of the Acceptable Ads Standard by developing an online telemetry pipeline to detect non-compliant ads. However, some limitations remain. First, our method relies on deterministic embedding of third-party resources to detect ads across two consecutive crawls. Although we wait for the `LOAD` event (indicating all resources have loaded), it does not ensure that all ads are in display mode, potentially resulting in an underestimation of violating ads.

Second, our work primarily focused on six key types of ads identified as unacceptable. While the Acceptable Ads Standard imposes strict limits on ad size, placement, and type, the complexities of measuring web elements restricted our ability to develop a fully comprehensive set of heuristics. The number of violations reported in our study may represent a lower bound.

Future research should aim to expand the set of heuristics to capture additional violations and improve the thoroughness of compliance checks. Furthermore, advancements in machine learning, particularly large language models (LLMs), could significantly enhance the detection of placement violations and help identify non-compliant ads across a broader range of contexts on the web.

## 9 CONCLUSION

In this paper, we assess the compliance of digital ads allowed under the Acceptable Ads Standard. We develop a crawling framework that identifies online ads and logs CSS properties during page crawls to detect non-compliant ads. Using this framework, we crawl 10,000 websites from the intersection of Tranco's top 100K and Acceptable Ads' Exception list. Despite these websites being allowed to show non-intrusive ads, our analysis finds that 1 in 10 display at least one forbidden ad format.

We identify two key issues in the Exception list that lead to these violations: overly permissive `^$document` rules and unblocking rules for offending elements. We demonstrate that improving the Exception list can reduce the number of violating domains by 32.4%. Overall, our findings highlight the need for continuous evaluation and enhancement of ad standards to better align with user expectations and improve the web experience.

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

## A   AD VIOLATIONS EXAMPLES

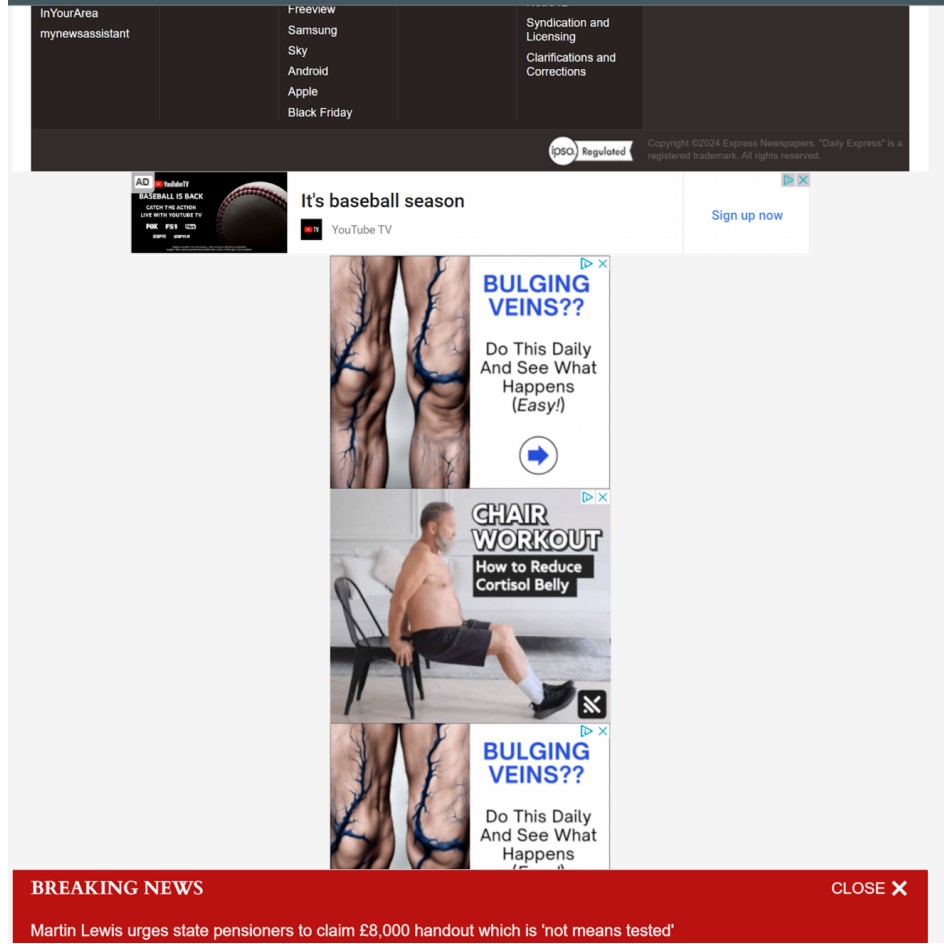

**Figure 7: Oversized Image Ad from express.co.uk**

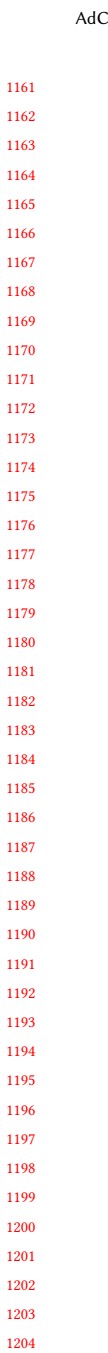

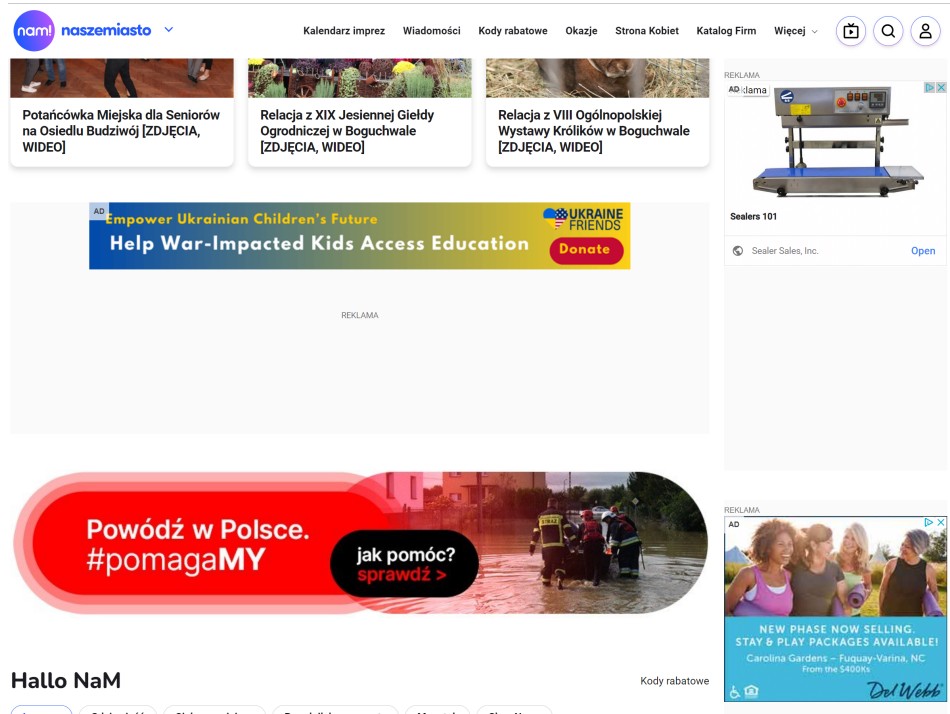

Figure 8: Overlay Image Ad from naszemiasto.pl

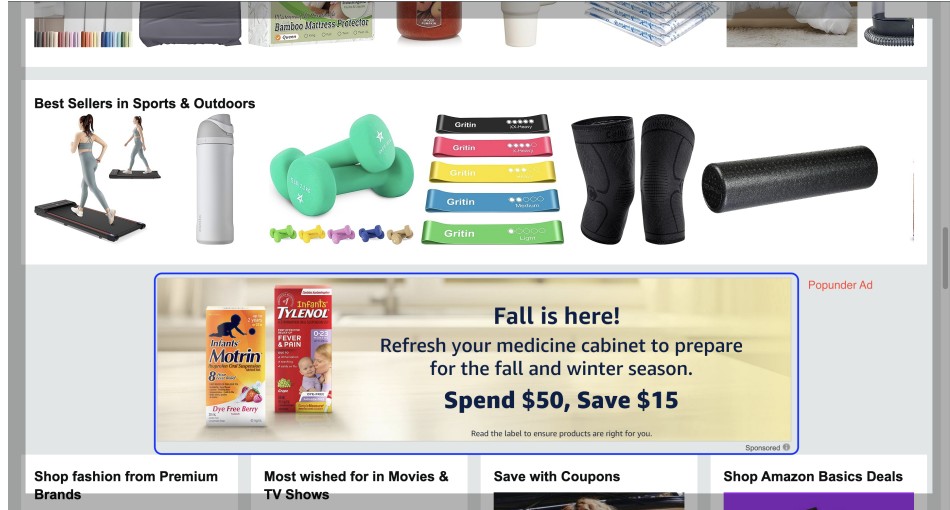

Figure 9: Popunder Ad from Amazon.ca

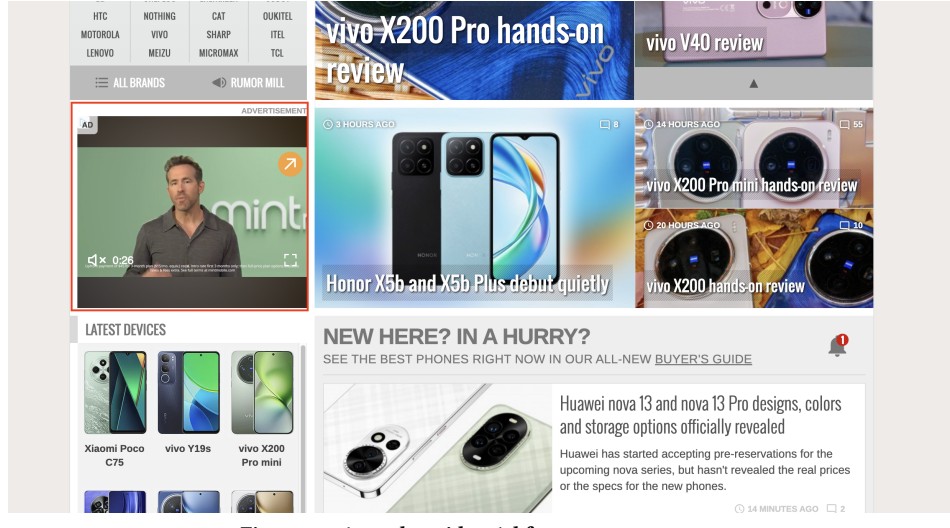

Figure 10: Autoplay video Ad from gsmarena.com

