# OpenReview forum: "Assessing Compliance in Digital Advertising: A Deep Dive into Acceptable Ads Standards"
_ACM.org/TheWebConf/2025/Conference — WWW 2025 Oral_

### Official Review · Reviewer_ZpTa · 2024-11-28

**Novelty:** 5
**Technical Quality:** 6

**Review:**

The paper presents an analysis of the adherence of websites to the Acceptable Ads Standard. It finds that in the observed sample, a substantial fraction of domains does violate these standards, despite being granted exceptions from ad blockers. The work presented in the paper is well motivated, clearly described and properly executed. While the results are not surprising, they are nevertheless important.

**Questions:**

Given the relatively short time in which the crawling was complete, why did the authors chose a comparatively small sample of all domains granted exceptions from block lists?

**Reviewer Confidence:**

3: The reviewer is confident but not certain that the evaluation is correct

**Scope:**

4: The work is relevant to the Web and to the track, and is of broad interest to the community

---

### Official Review · Reviewer_JU3d · 2024-12-02

**Novelty:** 5
**Technical Quality:** 6

**Review:**

The authors scrape 10k popular websites under different ad-blocking settings to measure the prevalence of ads in general, and intrusive ads in particular. I found the study design to be solid and well explained, the analysis appropriate, and the conclusions well explained.

I thoroughly enjoyed reading the paper and can only suggest small improvements in how the results are presented:

Figure 3 is not easy to read:
What is the main point of the figure? If it is to show that the violations are prevalent across ranks, the authors could just replace the figure with a Spearman correlation coefficient between the ad frequency and rank (and show that it’s not significant)
There is only one y-axis and it does not seem to correspond to any of the subplots. Why is it there at all?
The plots seem to be missing zeros (?). Instead, websites have a value imputed for them as somewhere between the two neighboring sites that do have ads. This is the most clear for Interstitial Ads and Pop-under ads, but present in all other plots.

Figure 5 is impossible to interpret. The values should be normalized by the prevalence of all ads, i.e: is Google the main purveyor of non-compliant ads because it’s particularly bad, or just because it’s the biggest ad network? This applies to Finding 3 as well. Further, the bands of 1-1000, 1001- … are of different sizes and we also know from Figure 3 that rank doesn’t matter, so why include this here.

Finding 4 - Table 3 shows that the prevalence of non-compliant ads dropped for each category. Perhaps it’s worth to interpret this fact as well.

Finding 6 - the authors show that their modified rule set increases recall. However, just like with every other binary classification problem, increasing recall will most likely lead to a decreased precision. The authors should report whether their ruleset leads to an increased prevalence of false positives (blocked, but compliant ads).

Finally, one ethical aspect. I am not flagging this for an ethics review, but it is important to mention in the paper:
- scraping websites while evading detection as automated bots means that ads are displayed to a non-legitimate user. Advertisers may be paying for each such instance. The authors should explain why they think the goal of the project outweighs the financial cost borne by the advertisers.

**Questions:**

Please provide the missing details on the precision/recall trade-off as described in the main review.

**Reviewer Confidence:**

2: The reviewer is willing to defend the evaluation, but it is likely that the reviewer did not understand parts of the paper

**Scope:**

4: The work is relevant to the Web and to the track, and is of broad interest to the community

---

### Official Review · Reviewer_BRLY · 2024-12-03

**Novelty:** 5
**Technical Quality:** 6

**Review:**

# Assessing Compliance in Digital Advertising: A Deep Dive into Acceptable Ads Standard

## Summary

This paper investigates the compliance of online advertisements with the Acceptable Ads Standards, which aim to strike a balance between user experience and monetisation. The authors focus on ads displayed on a subset of 10,000 domains, selected by intersecting Tranco’s top 100K domains with the Acceptable Ads exception list. Through a quantitative analysis, they assess the prevalence of non-compliant ads and evaluate the effectiveness of these standards. The findings reveal that approximately 10% of the websites in the exception list fail to comply with the standards, featuring at least one ad that violates guidelines for size, placement, or type.

In addition to identifying non-compliance rates, the paper highlights design flaws in the exception list that enable publishers to bypass the defined limitations. These flaws undermine the balance intended by the Acceptable Ads Standards, impacting user experience and reducing trust in the exception system. To address these shortcomings, the authors propose specific improvements to the exception list, aiming to prevent violations and enhance the effectiveness of the standards. By doing so, they provide actionable insights that could improve the user experience for millions of site visitors while ensuring fair monetization opportunities for publishers.
* * *

## Pros

This paper addresses a critical issue in online advertising—balancing monetisation needs with user experience through compliance with the Acceptable Ads Standards. The topic is both timely and relevant, as digital advertising continues to evolve amidst increasing consumer pushback against intrusive practices. The study is commendable for its empirical approach, utilising a large dataset of 10,000 domains derived from reputable sources like Tranco’s top 100K and the Acceptable Ads exception list. The methodology, which includes a quantitative analysis of compliance rates and ad formats, is robust and provides valuable insights into the prevalence of non-compliant ads. Moreover, the identification of design flaws in the exception list and the proposal of actionable improvements are practical contributions that could influence future implementations of ad standards.

* * *

## Cons

While the study is well-structured, there are a few areas where it could be improved:

- Experimental design: The design choices are generally reasonable, however, I expected a more robust approach towards various sources of ad-tracking. For instance, in section 4.1, the authors only focused on easylist, which is a reputable one, but other lists such as Fanboy are more extensive and detailed, especially on pop-ups, overlays and other patterns of advertising on the web.

- Another item is that the authors merely considered adBlock as their primary ad-blocker. This is not comprehensive as there are many other products which are different in performance and compliance with ad standards. For instance, the ad-block-by-design approach taken in browsers such as Brave turns out to be as effective (if not better) than ad-blocking extensions. What is the impact of such an approach and blocking on those platforms? This has not been discussed.

- Geolocation of the ads: This research has been predominantly conducted in the US. While the results are fine, I wonder what would be the impact of such blocking in locations where more constraints are imposed on ad-tracking. For instance, would the authors get similar results in the EU region where GDPR is in place? There is no discussion regarding such an approach in the result. The authors only briefly mention GDPR in the literature review but do not further discuss.

- Context and Impact: The paper could further contextualize the findings within the broader landscape of digital advertising. How significant is a 10% non-compliance rate in terms of revenue impact or user experience? What are the broader implications of these findings for advertisers and consumers?

- Proposed Improvements: The proposed improvements are briefly mentioned but not extensively detailed. A clearer roadmap or framework for implementing these changes, including potential challenges and trade-offs, would strengthen the paper's practical impact.


## Conclusion

The paper tackles an important aspect of online advertising and provides valuable insights into the compliance of digital ads with established standards. Its empirical approach and focus on practical recommendations are notable strengths. However, a deeper exploration of the methodology, broader contextual implications, and a more detailed discussion of proposed solutions would elevate its impact. The study lays a solid foundation for further research and practical implementation in this domain.

**Questions:**

- Why only easylist is included?
- What is the impact of such practices in ad-blocking browsers such as brave?
- What is the impact of these practices cross-region? e.g. the EU where GDPR is in place.

**Reviewer Confidence:**

3: The reviewer is confident but not certain that the evaluation is correct

**Scope:**

4: The work is relevant to the Web and to the track, and is of broad interest to the community

---

### Official Review · Reviewer_jNYo · 2024-12-03

**Novelty:** 4
**Technical Quality:** 3

**Review:**

This paper provides a detailed exploration of compliance with the Acceptable Ads Standard, using a novel two-crawl detection methodology to identify non-compliant ads on partner websites. The authors’ approach, leveraging telemetry data and a large dataset of high-traffic domains, effectively highlights the prevalence of violations, with 10% of websites displaying at least one non-compliant ad. While the methodology is rigorous, its reliance on specific heuristics and deterministic embedding of third-party resources limits its ability to detect the full range of potential violations.

The paper's clarity is reasonable, with well-structured sections and visualizations that aid understanding. However, the overemphasis on deterministic rules and the exclusion of mobile ads from the analysis constrain its practical applicability. Moreover, the ethical considerations section acknowledges minimal server impact, but there remains some unease about the broader implications of their telemetry approach, especially given the potential for underreporting violations due to incomplete ad rendering during crawls.

While the findings contribute to the conversation around ad compliance, the study’s impact is somewhat limited by its narrow scope and focus on desktop environments. The proposed improvements to the Exception list are promising but could benefit from real-world validation and a deeper exploration of their effectiveness in dynamic advertising ecosystems. Future research should address these gaps to provide a more comprehensive evaluation of compliance in digital advertising.

**Questions:**

Methodology for Ad Detection: How does your methodology account for dynamically rendered ads that may not load within the specified time window of the two-crawl process? Could this lead to underreporting of non-compliant ads?

Heuristic Design: Your heuristics for detecting non-compliant ads focus on specific formats. How did you decide on the thresholds (e.g., 80% screen size for oversized ads), and do you think these thresholds are too restrictive or lenient?

Scope of Ad Types: The study is limited to desktop ads. Do you anticipate significant differences in compliance rates or challenges if similar methodologies were applied to mobile ads, and why were mobile ads excluded?

Impact of Overly Permissive Rules: You identified overly permissive ^$document rules as a key contributor to non-compliance. How frequently do these rules originate from the Acceptable Ads Committee versus partner publishers? Could more centralized oversight address this issue?

Real-World Validation of Exception List Improvements: The proposed modifications to the Exception list reduced violations by 32.4%. Have you tested these changes in collaboration with stakeholders like the Acceptable Ads Committee or ad publishers to assess their real-world feasibility and impact?

**Reviewer Confidence:**

4: The reviewer is certain that the evaluation is correct and very familiar with the relevant literature

**Scope:**

4: The work is relevant to the Web and to the track, and is of broad interest to the community

---

### Official Review · Reviewer_Y586 · 2024-12-03

**Novelty:** 6
**Technical Quality:** 5

**Review:**

This paper concerns non-compliant digital advertising and the acceptable ads standard. The paper outlines a cohesive and well-executed measurement study using web crawling and quantitative analysis, with a number of interesting conclusions about the current standards and its allow-list practices.

The findings include that violating domains were present in about 10% of domains studied, and Oversized Image ads and Overlay ads were the most commonly occurring non-complying ads.

This paper represents a valuable contribution to understanding web advertising practices and compliance.

**Questions:**

Here are a few questions (but for the most part I think this work Is excellent)
Could you describe the possible ways that your methods for detecting non-compliant ads might "miss" in either direction? That is, either labeling OK ads as non-compliant or failing to label non-compliant ads as such? The Section 4.3 seems to be missing this sort of discussion.
The abstract has round numbers (10,000 out of 100,000) but it seems unlikely that the set you'd arrive at through intersecting sets would be 10,000 exactly. In Section 5.1 you arrive at the number 9,463. Could you clarify the number of domains and use it consistently?
Figures 3 and 5 need more explanation. In Figure 5, what does the width on left and right represent? Is it the ratio between these that you're after?

**Reviewer Confidence:**

3: The reviewer is confident but not certain that the evaluation is correct

**Scope:**

4: The work is relevant to the Web and to the track, and is of broad interest to the community